# Broca's area, variation and taxic diversity in early *Homo* from Koobi Fora (Kenya)

Amélie Beaudet[1,2,3]*, Edwin de Jager[1]

[1]Laboratoire de Paléontologie, Évolution, Paléoécosystèmes et Paléoprimatologie (PALEVOPRIM), UMR 7262 CNRS & University of Poitiers, Poitiers, France; [2]Department of Archaeology, University of Cambridge, Cambridge, United Kingdom; [3]School of Geography, Archaeology and Environmental Studies, University of the Witwatersrand, Johannesburg, South Africa

**Abstract** Because brain tissues rarely fossilize, pinpointing when and how modern human cerebral traits emerged in the hominin lineage is particularly challenging. The fragmentary nature of the fossil material, coupled with the difficulty of characterizing such a complex organ, has been the source of long-standing debates. Prominent among them are the uncertainties around the derived or primitive state of the brain organization in the earliest representatives of the genus *Homo*, more particularly in key regions such as the Broca's area. By revisiting a particularly well-preserved fossil endocast from the Turkana basin (Kenya), here we confirm that early *Homo* in Africa had a primitive organization of the Broca's area ca. 1.9 million years ago. Additionally, our description of KNM-ER 3732 adds further information about the variation pattern of the inferior frontal gyrus in fossil hominins, with implications for early *Homo* taxic diversity (i.e. one or two *Homo* species at Koobi Fora) and the nature of the mechanisms involved in the emergence of derived cerebral traits.

## eLife assessment

This **important** study uses the brain endocast of a ~1.9-million-year-old hominin fossil from Kenya, attributed to genus *Homo*, to show that the organization of the Broca's area in members of early *Homo* was primitive. Specifically, the prefrontal sulcal pattern in this early *Homo* specimen more closely resembles that of chimpanzees than of modern humans. Because Broca's area is associated with speech function, the **compelling** evidence from this study is relevant for understanding the timing and trajectory of evolution of speech related traits in our genus. Coupled with its potential implications for taxonomic classification, this study will be of interest to paleoanthropologists, paleontologists, archaeologists, and neuroscientists.

## Introduction

The modern human brain is an exceptionally complex, highly specialized, and extremely costly machinery. Because of the fragmentary nature of the hominin fossil record, assessing when and how changes in the brain of our ancestors happened, and inferring any related functional, behavioral, and metabolic consequences, is particularly challenging (*Zollikofer and De León, 2013*). Nonetheless, reconstructing the chronological and taxonomical context of the emergence of derived cerebral traits is a prerequisite for disentangling underlying evolutionary processes. For instance, local reorganization of specific areas in an overall primitive hominin brain would support a mosaic-like evolutionary pattern (e.g. *Holloway et al., 2004*), and raise essential questions on the role of selection pressure (or absence of; rev. in *Beaudet, 2021*). Beyond the value of such information on the origins of the human

*For correspondence:
beaudet.amelie@gmail.com

**Competing interest:** The authors declare that no competing interests exist.

brain, the assumption that *Homo* developed a uniquely complex brain organization (e.g. *Tobias, 1987*) requires further evidence.

Although brains rarely fossilize, it is possible to glean structural information about the evolutionary history of the hominin brain by studying sulcal imprints in fossil brain endocasts (*Neubauer, 2014*). In this regard, the Broca's area has been the focus of much interest in paleoneurology due to striking structural differences between extant human and chimpanzee brains and endocasts, and the implication of this area in articulated language (rev. in *Beaudet, 2017*). *Ponce de León et al., 2021* thoroughly examined brain endocasts of *Homo* specimens in eastern Africa and Eurasia and demonstrated that the organization of the Broca's area in the earliest representatives of the genus before 1.5 Ma was primitive. Because the imprints of the Broca's cap in endocasts are not always readable, they used the coronal suture and surrounding sulcal imprints as a proxy to identify frontal lobe expansion (i.e. derived condition). Unfortunately, in some of the oldest and key specimens from Africa that could represent the >1.5 Ma condition (e.g. KNM-ER 1470) the interpretations remained inconclusive.

Testing the hypothesis of *Ponce de León et al., 2021* of a primitive brain in the earliest representatives of the genus *Homo* before 1.5 Ma thus necessitates (i) an excellent preservation of very fine neuroanatomical details in fossil endocasts, and (ii) reliable information on their taxonomic identity (i.e. *Homo*) and stratigraphic context (i.e. before/after 1.5 Ma). The hominin specimen KNM-ER 3732 from Kenya, which fulfills these requirements, has the potential to shed new light on this conundrum. KNM-ER 3732 was discovered during the 1974–1975 field program in the Koobi Fora Formation (area 115), east of Lake Turkana in Kenya (*Leakey, 1976*; *Wood, 1991*). KNM-ER 3732 consists of a calotte, left zygoma, and a natural endocast. The expanded neurocranium and robust upper face support an attribution to *Homo* sp. indet. (*Wood, 1991*). The specimen was lying below the KBS Tuff of the upper Burgi Member that is dated to 1.87 million years ago, which provides a minimum age (*Feibel et al., 1983*; *Feibel et al., 2006*). Accordingly, if KNM-ER 3732 shows a derived cerebral condition, the hypothesis of >1.5 Ma *Homo* being associated with a primitive organization of the Broca's area is falsified. Here, we provide a comparative study of the natural endocast of the pre-1.5 Ma *Homo*

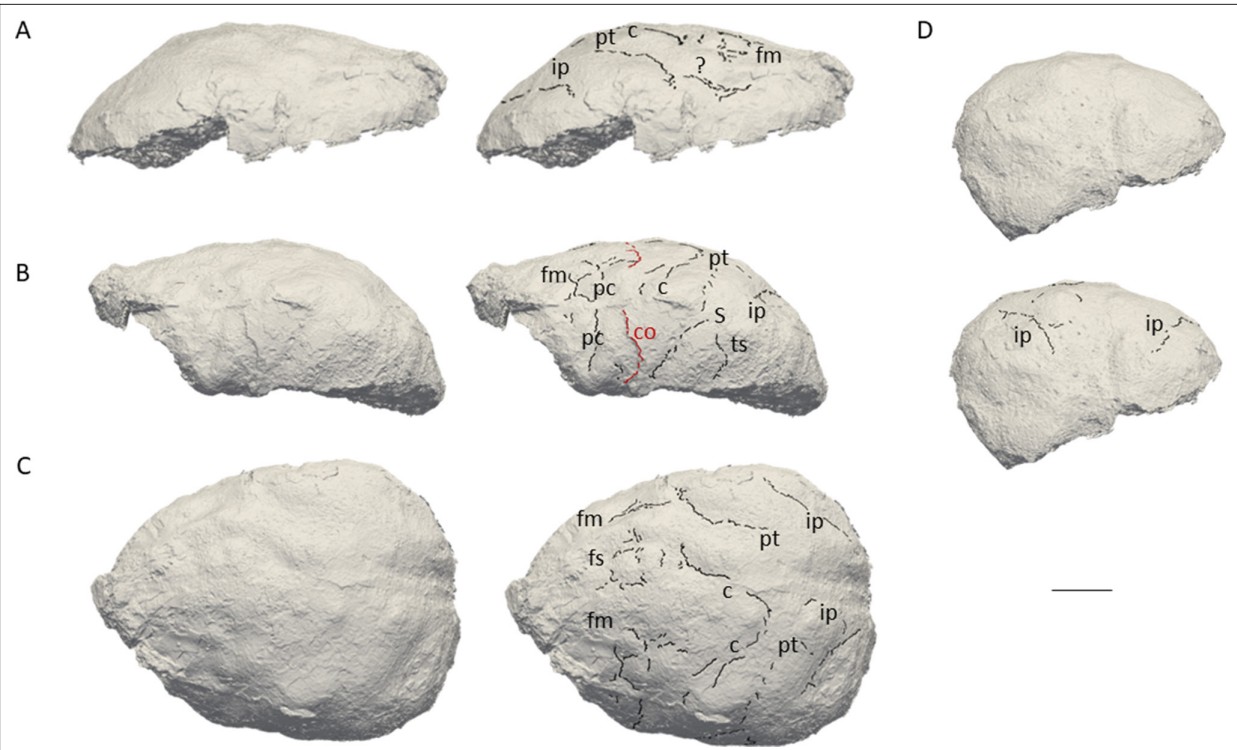

**Figure 1.** 3D virtual rendering of the natural endocast of KNM-ER 3732 and identification of sulcal imprints. KNM-ER 3732 is shown in the lateral right (**A**), lateral left (**B**), dorsal (**C**), and posterior (**D**) views. ar: ascending ramus of the lateral fissure; c: central sulcus; CO: coronal suture; fi: inferior frontal sulcus; fm: middle frontal sulcus; ip: intra-parietal sulcus; pc: pre-central sulcus; pt: post-central sulcus. Scale bar: 2 cm.

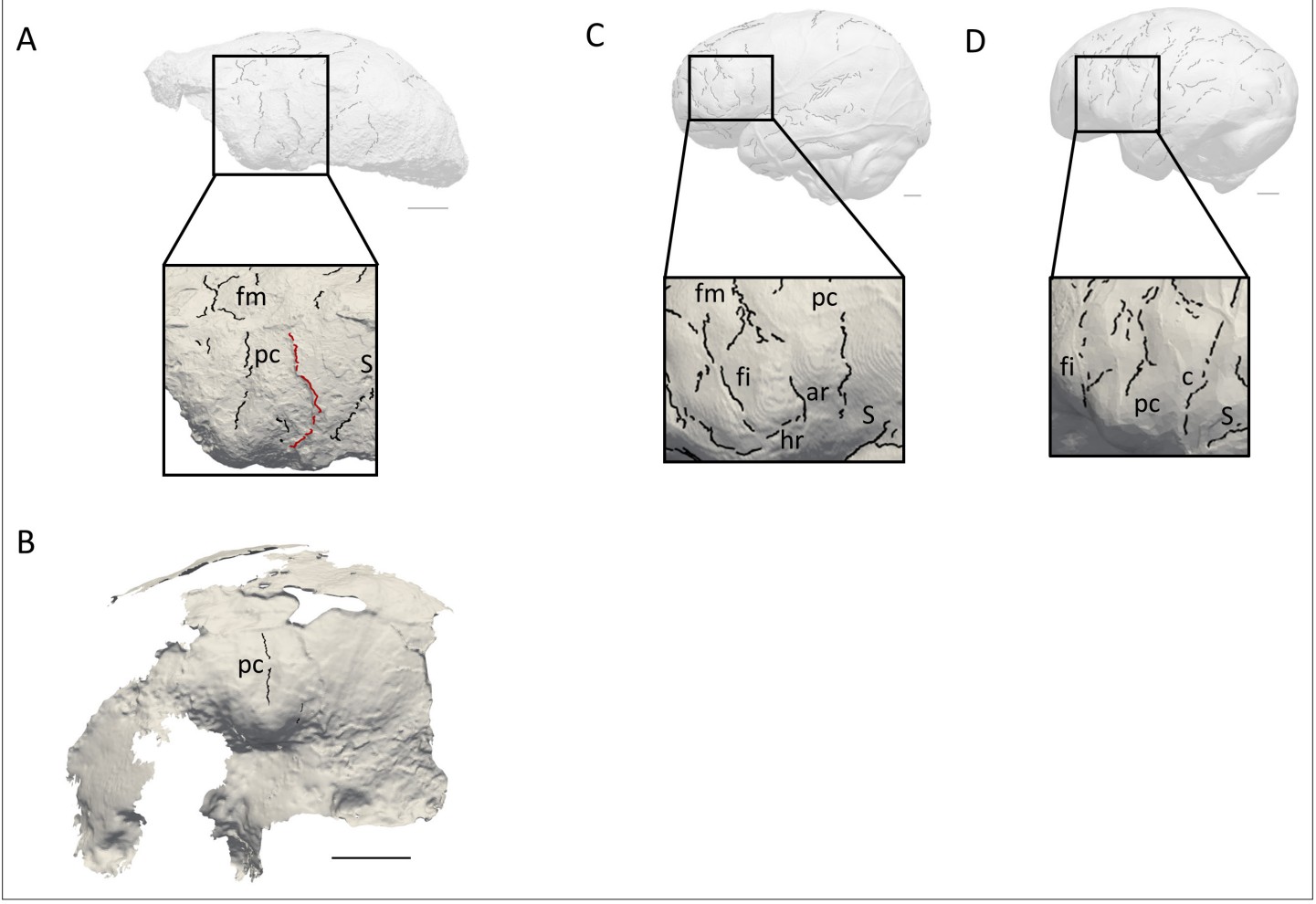

**Figure 2.** Comparison of the sulcal patterns identified in the inferior frontal area of KNM-ER 3732. The natural endocast (**A**) and braincase (**B**) of KNM-ER 3732 are compared to the virtual endocasts of extant human (**C**) and chimpanzee (**D**) individuals. Images not to scale. ar: ascending ramus of the lateral fissure; c: central sulcus; co: coronal suture; fi: inferior frontal sulcus; fm: middle frontal sulcus; ip: intra-parietal sulcus; pc: pre-central sulcus; pt: post-central sulcus. Scale bar: 2 cm.

specimen KNM-ER 3732 to test the hypothesis of a late emergence of a modern Broca's area in the hominin lineage.

## Results

### KNM-ER 3732 offers a glimpse of early *Homo* prefrontal organization

The natural endocast of KNM-ER 3732, initially described by *Holloway et al., 2004*, preserves the dorsal part but misses the frontal pole, the occipital and temporal lobes, as well as the entire ventral surface (*Figure 1*). *Holloway et al., 2004* estimated the endocranial volume as about 750–800 cc but pointed out the low reliability of their estimate. *Wood, 1991* (Page 132) identified an 'irregular bony excrescence on the surface of the right parietal at the level of lambda and 32 mm from the midline. It projects 6 mm from the surface of the bone, and may be an example of myositis ossificans associated with damage to the overlying temporal muscle.' Because of the location (right parietal) and nature (traumatic) of the outgrowth, it does not affect our area of study (i.e. left frontal bone) (*Walczak et al., 2015*). The gyral and sulcal details are well-preserved, particularly in the prefrontal region. The superior, middle, and inferior frontal sulci are visible on both hemispheres. On the left hemisphere, the Broca's cap is prominent. A vertical groove, identified as the precentral sulcus, separates the Broca's cap in half (*Figure 2A*). On both hemispheres, the central sulcus intersects the inter-hemispheric

scissure and the post-central sulcus seems to be connected to the lateral fissure. The intra-parietal sulcus can be found in the parieto-occipital region of both hemispheres. The examination of the internal surface of the braincase (*Figure 2B*) confirms the pattern described on the natural endocast.

## KNM-ER 3732 has a primitive prefrontal cortex

When compared to extant human and chimpanzee sulcal patterns, KNM-ER 3732 closely resembles the latter (*Figure 2C–D*). In extant human brains and endocasts, the inferior frontal sulcus often transects the Broca's cap, while the ascending ramus of the lateral fissure caudally borders the prominence (*Figure 2C–D*; *Connolly, 1950*; *de Jager et al., 2019*; *de Jager et al., 2022*). In chimpanzees, the central sulcus is placed more rostrally, and the inferior portion of the precentral sulcus bisects the Broca's cap that is bordered rostrally by the inferior frontal sulcus (*Figure 2E–F*; *Connolly, 1950*; *Falk et al., 2018*). The coronal suture runs in between the precentral and central sulci in KNM-ER 3732, which points towards the primitive configuration described in *Ponce de León et al., 2021*. The sulcal pattern seen in the prefrontal cortex of KNM-ER 3732, that was detected in both the natural endocast and the inner surface of the cranium (*Figure 2A–B*), and more particularly the deeply marked precentral sulcus that incised the Broca's cap, is not found in any of the early *Homo* specimens described in *Ponce de León et al., 2021* but approximates the condition seen in the Dmanisi cranium D2282 (Fig. S1B in *Ponce de León et al., 2021*). However, this pattern is seen in a contemporaneous non-*Homo* hominin specimen in South Africa, i.e. *Australopithecus sediba* (*Falk, 2014*).

## Discussion

### Paleoneuroanatomy supports taxic diversity within early *Homo*

Overall, the present study not only demonstrates that Ponce de León et al.'s (2021) hypothesis of a primitive brain of early *Homo* cannot be rejected, but also adds information about the variation pattern of the inferior frontal gyrus. In particular, the diversity of the prefrontal sulcal patterns of hominin endocasts at Koobi Fora revives the debate about the possible presence of two early *Homo* species in this locality (i.e. *Homo habilis sensu stricto* and *Homo rudolfensis*). In 1983, *Falk, 1983* published the description of the endocasts of two *Homo* specimens from Koobi Fora, KNM-ER 1470 and KNM-ER 1805. Her analysis supported the co-existence of two morphs, KNM-ER 1470 representing a more derived human-like sulcal pattern. Interestingly, cerebral evidence brought up by her analysis matched other studies that emphasized the more derived craniodental anatomy of specimens attributed to *Homo rudolfensis* as opposed to the more primitive (*Australopithecus*-like) traits identified within *Homo habilis sensu stricto* (*Leakey et al., 2012*). While describing the external morphology of the neurocranium, (*Leakey, 1976*: 575) noted that KNM-ER 3732 was 'strikingly similar to KNM-ER 1470.' This resemblance is not reflected in their cerebral organization since the present study rather suggests a primitive organization of the Broca's cap in KNM-ER 3732. If KNM-ER 1470 had indeed a derived brain, taxic diversity as a source of variation cannot be discarded. If we go further down that route, the similarities between KNM-ER 3732 and *Australopithecus sediba* suggested by our study could be an argument supporting the presence of *Australopithecus* in Koobi Fora or the absence of a definite threshold between the two genera based on the morphoarchitecture of their endocasts (*Wood and Collard, 1999*).

### The evolutionary history of the human Broca's area unraveled

Beyond the taxonomic aspect, variation detected in Koobi Fora could provide information about underlying evolutionary mechanisms, and more specifically the process of fixation of an adaptive variant, i.e. a new organization of the Broca's area and the increase of neural interconnectivity in this region (*Essen, 1997*). Such neurological changes might have had deep implications for the emergence of novel behaviors, such as articulated language (*Beaudet, 2017*; *Beaudet, 2021*). Within this scenario, the study from *Ponce de León et al., 2021* would suggest that this trait became stabilized by 1.5 million years ago. The identification of protracted brain growth as early as in *Australopithecus afarensis* (*Gunz et al., 2020*), inducing longer exposure to the social environment during brain maturation (rev. in *Hublin et al., 2015*), could be consistent with a derived organization of the Broca's area being selected as a response to social environmental stimuli through developmental plasticity,

culminating in this variant becoming dominant within *Homo*. In parallel, the possibility of allometric scaling and the influence of brain size on sulcal patterns in early *Homo* has to be further explored.

## Materials and methods

### Materials

KNM-ER 3732 is currently housed in the National Museums of Kenya in Nairobi (Kenya). We used brain and endocast atlases published in *Connolly, 1950*, *Falk et al., 2018*, and *de Jager et al., 2019*; *de Jager et al., 2022*; see also https://www.endomap.org/ for comparing the pattern identified in KNM-ER 3732 to those described in extant humans and chimpanzees. To the best of our knowledge, these atlases are the most extensive atlases of extant human and chimpanzee brains/endocasts available to date and are widely used in the literature to explore variability in sulcal patterns. In *Figure 2*, the extant human and chimpanzee conditions are illustrated by one extant human (adult female) and one extant chimpanzee (adult female) specimens from the Pretoria Bone Collection at the University of Pretoria (South Africa) and from the Royal Museum for Central Africa in Tervuren (Belgium), respectively (*Beaudet et al., 2018*).

### Digitization

Both the natural endocast and the cranium of KNM-ER 3732 were scanned using an Artec Space Spider 3D scanner and reconstructed with the software Artec Studio 16 X. The 3D mesh can be viewed on MorphoSource. The comparative specimens were imaged by microfocus X-ray tomography (*Beaudet et al., 2018*) at the South African Nuclear Energy Corporation in Pelindaba (South Africa) and at the Centre for X-ray Tomography of Ghent University (UGCT) in Ghent (Belgium). Virtual endocasts were generated using Endex software (*Subsol et al., 2010*).

### Detection and identification of sulcal imprints

Sulcal imprints were automatically detected through a geometry-based method using curvature lines computed from the natural endocast and the inner surface of the cranium. Sulcal imprints, considered as variation points of the surface on a triangle mesh, were detected through a geometry-based method using curvature lines defined as salient subsets of the extrema of the principal curvatures on surfaces (*Yoshizawa et al., 2008*; *Beaudet et al., 2016*). Vascular imprints and non-anatomical structures (e.g. fractures) were manually removed through a customized script written in MATLAB R2013a (Mathworks) that is available online (*Dumoncel, 2019*; https://gitlab.com/jeandumoncel/curve-editor).

## Acknowledgements

We thank Emmanuel Ndiema and the curatorial staff of the National Museums of Kenya for collection access. For scientific/technical discussion, we are grateful to: G Castelli (Cambridge), M Mirazón-Lahr (Cambridge), R Holloway (New York), F Spoor (London). AB is funded by the National Research Foundation of South Africa (Research Development Grants for Y-Rated Researchers, grant number 129336), the University of Cambridge, and the Centre National de la Recherche Scientifique. EdJ is funded by the University of Cambridge Harding Distinguished Postgraduate Scholars Programme. AB and EdJ are funded by the South Africa/France (PROTEA) Joint Research Programme (grant number 129923) and the McDonald Institute for Archaeological Research. We are grateful to the three reviewers and the editors for their comments and suggestions.

## Additional information

### Funding

| Funder | Grant reference number | Author |
| --- | --- | --- |
| National Research Foundation | 129336 | Amélie Beaudet |
| University of Cambridge | | Amélie Beaudet |

| Funder | Grant reference number | Author |
| --- | --- | --- |
| Centre National de la Recherche Scientifique | CPJ-Hominines | Amélie Beaudet |
| University of Cambridge | Harding Distinguished Postgraduate Scholars Programme | Edwin de Jager |
| South Africa/France Joint Research Programme | 129923 | Amélie Beaudet |
| McDonald Institute for Archaeological Research | | Amélie Beaudet |

The funders had no role in study design, data collection and interpretation, or the decision to submit the work for publication.

### Author contributions

Amélie Beaudet, Conceptualization, Formal analysis, Funding acquisition, Investigation, Methodology, Writing - original draft; Edwin de Jager, Formal analysis, Validation, Writing - review and editing

### Author ORCIDs
Amélie Beaudet http://orcid.org/0000-0002-9363-5966
Edwin de Jager http://orcid.org/0000-0003-3199-8566

### Ethics

Ethical clearance for the use of extant human cranium was obtained from the Main Research Ethics committee of the Faculty of Health Sciences, University of Pretoria in February 2016.

Reviewer #1 (Public Review): https://doi.org/10.7554/eLife.89054.3.sa1
Reviewer #2 (Public Review): https://doi.org/10.7554/eLife.89054.3.sa2
Reviewer #3 (Public Review): https://doi.org/10.7554/eLife.89054.3.sa3
Author Response https://doi.org/10.7554/eLife.89054.3.sa4

## Additional files

### Supplementary files
• MDAR checklist

### Data availability

The 3D mesh of KNM-ER 3732 can be viewed on MorphoSource (https://www.morphosource.org/concern/media/000497752?locale=en). 3D models of comparative endocasts can be made available upon request if permission from the respective curators is granted. Indeed, extant human and chimpanzee endocasts derive from dry crania that are curated in osteological collections with controlled access. Ethics clearance must be obtained in the case of the human endocast. As such, an application should be submitted and approved by the curators. For the extant human endocast, permission should be requested from the Pretoria Bone Collection at the University of Pretoria, South Africa (Ericka L'Abbé, ericka.labbe@up.ac.za). For the extant chimpanzee endocasts, permission should be requested from the Royal Museum for Central Africa, Belgium (Emmanuel Gilissen, emmanuel.gilissen@africamuseum.be). If approved, the first author will share the 3D models of the endocasts. The customized script written in MATLAB R2013a (Mathworks) for editing detected curves is available online (https://gitlab.com/jeandumoncel/curve-editor).

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
