## [Editor Report · eLife assessment]

This **important** study uses the brain endocast of a ~1.9-million-year-old hominin fossil from Kenya, attributed to genus *Homo*, to show that the organization of the Broca's area in members of early *Homo* was primitive. Specifically, the prefrontal sulcal pattern in this early *Homo* specimen more closely resembles that of chimpanzees than of modern humans. Because Broca's area is associated with speech function, the **compelling** evidence from this study is relevant for understanding the timing and trajectory of evolution of speech related traits in our genus. Coupled with its potential implications for taxonomic classification, this study will be of interest to paleoanthropologists, paleontologists, archaeologists, and neuroscientists.

---

## [Referee Report · Reviewer #1 (Public Review)]

The cerebral cortex, or surface of the brain, is where humans do most of their conscious thinking. In humans, the grooves (sulci) and bumps (convolutions) have a particular pattern in a region of the frontal lobe called Broca's area, which is important for language. Specialists study features imprinted on the internal surfaces of braincases in early hominins by casting their interiors, which produces so-called endocasts. A major question about hominin brain evolution concerns when, where, and in which fossils a humanlike Broca's area first emerged, the answer to which may have implications for the emergence of language. The researchers used advanced imaging technology to study the endocast of a hominin (KNM-ER 3732) that lived about 1.9 million years ago (Ma) in Kenya to test a recently published hypothesis that Broca's remained primitive (apelike) prior to around 1.5 Ma. The results are consistent with the hypothesis and raise new questions about whether endocasts can be used to identify the genus and/or species of fossils.

---

## [Referee Report · Reviewer #2 (Public Review)]

The authors present new data of endocranial surface details from the early Homo specimen KNM-ER 3732 and discuss the evolution of brain surface features that might be related to the evolution of language in the hominin lineage.

Comments and issues raised by the reviewers have been addressed adequately. I am sure that this contribution will revive discussion about these issues.

---

## [Referee Report · Reviewer #3 (Public Review)]

The authors provide a detailed analysis of the sulcal and sutural imprints preserved on the natural endocast and associated cranial vault fragments of the KNM-ER3732 early Homo specimen. The analyses indicate a primitive ape-like organization of this specimen's frontal cortex. Given the geological age of around 1.9 million years, this is the earliest well-documented evidence of a primitive brain organization in African Homo.

The various points raised by the reviewers and the responses provided by the authors illustrate that paleoneurology is a research field where little consensus has been reached over the past century. This is due not only to the fragmentary preservation of most fossil endocasts, but also to the limitations of scientific inference in general, and paleoneurological inference in particular. Like any scientific hypothesis, a paleoneurological hypothesis cannot be proven, but at best be falsified, leaving a wide field of possible alternative hypotheses. Furthermore, endocranial morphology does not equate cerebral morphology. A classical example: the endocranial Broca cap is not identical to the cortical Broca area. And last but not least, taxonomy cannot resolve questions of phylogeny.

---

## [Author Response]

The following is the authors’ response to the original reviews.

**Reviewer #1 (Public Review):**
The cerebral cortex, or surface of the brain, is where humans do most of their conscious thinking. In humans, the grooves (sulci) and bumps (convolutions) have a particular pattern in a region of the frontal lobe called Broca's area, which is important for language. Specialists study features imprinted on the internal surfaces of braincases in early hominins by casting their interiors, which produces so-called endocasts. A major question about hominin brain evolution concerns when, where, and in which fossils a humanlike Broca's area first emerged, the answer to which may have implications for the emergence of language. The researchers used advanced imaging technology to study the endocast of a hominin (KNM-ER 3732) that lived about 1.9 million years ago (Ma) in Kenya to test a recently published hypothesis that Broca's remained primitive (apelike) prior to around 1.5 Ma. The results are consistent with the hypothesis and raise new questions about whether endocasts can be used to identify the genus and/or species of fossils.

We would like to thank Rev. 1 for their comments on our paper.

**Reviewer #2 (Public Review):**
The authors tried to support the hypothesis that early Homo still had a primitive condition of Broca's cap (the region in fossil endocasts corresponding to Broca's area in the brain), being more similar to the condition in chimpanzees than in humans. The evidence from the described individual points to this direction but there are some flaws in the argumentation.

We are grateful to Rev. 2 for their comments, although we partially agree with some of them.

First, we would like to rectify the statement of Rev. 2 that we “tried to support the hypothesis that early Homo still had a primitive condition of Broca's cap”, indeed, our aim was to test this hypothesis and not to try to validate it.

First, only one human and one chimpanzee were used for comparison, although we know that patterns of brain convolutions (and in addition how they leave imprints in the endocranial bones) are very variable.

We understand the point raised by Rev. 2 about the variation of brain convolutions in humans and chimpanzees. We used atlases published by Connolly (1950), Falk et al. (2018) and de Jager et al. (2019, 2022) to analyse the endocast of KNM-ER 3732 and compare it to the extant human and chimpanzee cerebral conditions. However, in Figure 2, for the sake of clarity only two Homo and Pan specimens were used to illustrate the comparison (as it has been done in other published papers, e.g., Carlson et al., 2011; Science, Gunz et al., 2020 Sci Adv). In the revised version, we modified the manuscript to explain further our approach (line 156) “We used brain and endocast atlases published in Connolly (1950), Falk et al. (2018) and de Jager et al. (2019, 2022; see also www.endomap.org) for comparing the pattern identified in KNM-ER 3732 to those described in extant humans and chimpanzees. To the best of our knowledge, these atlases are the most extensive atlases of extant human and chimpanzee brains/endocasts available to date and are widely used in the literature to explore variability in sulcal patterns. In Figure 2, the extant human and chimpanzee conditions are illustrated by one extant human (adult female) and one extant chimpanzee (adult female) specimens from the Pretoria Bone Collection at the University of Pretoria (South Africa) and in the Royal Museum for Central Africa in Tervuren (Belgium), respectively (Beaudet et al., 2018).”.

Second, the evidence from this fossil specimen adds to the evidence of previously describe individuals but still not yet fully prove the hypothesis.

We tempered our discussion by concluding that (line 116) “Overall, the present study not only demonstrates that Ponce de León et al.’s (2021) hypothesis of a primitive brain of early Homo cannot be rejected, but also adds information […]”.

Third, there is a vicious circle in using primitive and derived features to define a fossil species and then using (the same or different) features to argue that one feature is primitive or derived in a given species. In this case, we expect members of early Homo to be derived compared to their predecessors of the genus Australopithecus and that's why it seems intriguing and/or surprising to argue that early Homo has primitive features. However, we should expect that there is some kind of continuum or mosaic in a time in which a genus "evolves into" another genus. This discussion requires far more discussions about the concepts we use, maybe less discussion about what is different between the two groups but more discussion about the evolutionary processes behind them.

We fully agree with Rev. 2 on this aspect. We believe that identifying these differences/similarities between fossil and extant hominids constitute the first step of a better understanding of the evolutionary mechanisms. Our work suggests indeed a certain continuity between genera and raises questions on the genus concept and how to interpret the specimens currently attributed to early Homo. In the revised version of the manuscript we included a reference to this possible scenario (line 134): “[…] or to the absence of a definite threshold between the two genera based on the morphoarchitecture of their endocasts (Wood and Collard, 1999).”.

Fourth, the data of convolutional imprints presented are rather subjective when identifying which impressions represent which brain convolutions. Not seeing an impression does not necessarily mean that the corresponding brain feature did not exist. Interestingly, the manuscript does not mention and discuss at all the frontoorbital sulcus. This is a sulcus that usually runs from the orbital surface of the frontal lobe up to divide the inferior frontal gyrus in chimpanzees, a condition totally different than in humans who do not have a frontoorbital sulcus. Could such a sulcus be identified, this would provide a far more convincing argument for a primitive condition in this specimen. In Australopithecus sediba, e.g., the condition in this region seems to be a mosaic in which some aspects of the morphology seem to be more modern while one of the sulcual impressions can well be interpreted as a short frontoorbital sulcus. For this specimen, by the way, I would come back to my third point above: some experts in the field might argue that this specimen could belong to Homo rather than Australopithecus...

We agree that the presence of a fronto-orbital sulcus would be more conclusive. However, this sulcus has not been identified in KNM-ER3732 and the region in which we would expect to find it is not preserved. As demonstrated by Ponce de León et al. (2021), because of the topographic relationships between sulci (and cranial structures), it is possible to interpret imprints on endocasts and the evolutionary polarity of some traits even in the absence of landmarks such as the fronto-orbital sulcus. In Australopithecus sediba the main derived feature of the endocast corresponds to the ventrolateral bulge in the left inferior frontal gyrus, and not to the sulcal pattern itself (Carlson et al., 2011 Science). However, the discussion around the taxonomic status of this taxon confirms the urgent need for reconsidering specimens from that time period and clarifying the mosaic-like or concerted evolution of the derived Homo-like traits within our lineage. Regarding the subjective nature of this approach, we invite readers to examine the specimen on MorphoSource (https://www.morphosource.org/concern/media/000497752?locale=en) and to request access to the National Museums of Kenya to the physical or virtual specimen to falsify our hypothesis.

According to my arguments above, I think that this manuscript might revive interesting discussions about this topic but it is not likely to settle them because the data presented are not strong enough to fully support the hypothesis.

We would be more than happy to consider new/other specimens with similar chronological and geographical contexts and investigate further this hypothesis in the future.

**Reviewer #3 (Public Review):**
The authors provide a detailed analysis of the sulcal and sutural imprints preserved on the natural endocast and associated cranial vault fragments of the KNM-ER3732 early Homo specimen. The analyses indicate a primitive ape-like organization of this specimen's frontal cortex. Given the geological age of around 1.9 million years, this is the earliest well-documented evidence of a primitive brain organization in African Homo.In the discussion, the authors re-assess one of the central questions regarding the evolution of early Homo: was there species diversity, and if yes, how can we ascertain it? The specimen KNM-ER1470 has assumed a central role in this debate because it purportedly shows a more advanced organization of the frontal cortex compared to other largely coeval specimens (Falk, 1983). However, as outlined in Ponce de León et al. 2021 (Supplementary Materials), the imprints on the ER1470 endocranium are unlikely to represent sulcal structures and are more likely to reflect taphonomic fracturing and distortion. Dean Falk, the author of the 1983 study, basically shares this view (personal communication). Overall, I agree with the authors that the hypothesis to be tested is the following: did early Homo populations with primitive versus derived frontal lobe organizations coexist in Africa, and did they represent distinct species?I greatly appreciate that the authors make available the 3D surface data of this interesting endocast.

We are grateful to Rev. 3 for their comments and for contextualizing our finding. We would also like to point out that, although the 3D surface can be viewed on MorphoSource, permission from the National Museums of Kenya has to be requested for studying the specimen and getting access to the physical specimen and/or the 3D model.

**Reviewer #1 (Recommendations For The Authors):**
Holloway, Broadfield & Yuan (2004) estimate ER 3732 as having a cranial capacity of 750 cc, which is larger than chimps and australopiths and similar to ER 1470 (752 cc, same reference). (That for Dmanisi 2282 is somewhat smaller at around 650 cc.) Cranial capacities should be mentioned along with added discussion about possible allometric scaling of (increased) numbers of sulci with increasing brain size as well as possible shifts in locations of sulci relative to cranial sutures in larger-brained (including due to ontogenetic maturation) in individuals/species. Could these variables (especially brain size) be relevant for your discussion/conclusions?

We thank Rev. 1 for their suggestion. We included the estimate by Holloway et al. (2004) (line 95): “Holloway et al. (2004) estimated the endocranial volume as about 750-800 cc but insisted on the low reliability of their estimate.”. Additionally, we raised the possibility of potential allometric effect (line 149): “In parallel, the possibility of allometric scaling and influence of brain size on sulcal patterns in early Homo has to be further explored.” for future discussion.

From the two figures, it appears that the authors produced a virtual endocast from the cranial remains of ER 3732 and compared its features with those seen on a virtual reproduction of the corresponding natural endocast. If so, this needs to be clarified in the text, not just the figures.

We thank Rev. 1 for their suggestions that were integrated.

**Reviewer #3 (Recommendations For The Authors):**
While the sulcal imprints on the left hemisphere can be interpreted unambiguously, the anatomical assignment of those on the right side may need to be reconsidered, as they are more ambiguous. For example, the postcentral sulcus (pt) almost touches the middle frontal sulcus, which is an unlikely natural configuration.

We agree that the configuration on the right hemisphere is intriguing, especially when compared to the extant human and chimpanzee atlases. As such, we decided to change the label for what we think could be the inferior frontal sulcus and leave a question mark instead.

I encourage the authors to include:A posterior view in Figure 1, and mark the lambdoid suture, parts of which seem to be preserved especially on the left side. This will help the readership to better understand which parts of the endocranial morphology are preserved.A scale bar would be of great utility to appreciate the small size of this specimen. The distance from bregma to the Broca cap seems to be short, indicating an endocranial volume much smaller than the published estimate of 750 ccm. Perhaps the authors can provide a new estimate, which would provide further support for the arguments proposed in the discussion section, especially the question of any presence of Australopithecus at Koobi Fora.

We included a posterior view of the specimen in Figure 1 and scale bar and modified the legend accordingly. Unfortunately, we were not able to identify with certainty the feature that could correspond to the lambdoid suture. We might see the impression where the parietal bone meets the occipital bone, but there is a risk of misidentification (which is an issue frequently raised in the literature, see for example Gunz et al. 2020 Sci Adv). Concerning the endocranial volume, in the revised version of the manuscript we included the estimate by Holloway et al. (2004). Because the specimen only preserves the superior part, we are reluctant in providing an estimate of the total volume. However, we agree that this would be an interesting feature to integrate in the interpretation of this specimen.

Minor pointsThis sentence needs to be clarified: «The superior temporal sulcus nearly intersects the lateral fissure on the right hemisphere».The terms «Broca's region» and «orbital cap» need some more context. Do the authors mean «Broca's cap» in either instance?

We clarified/modified when needed, thank you very much.

We included minor corrections in addition to those recommended by the reviewers:

-Lines 50, 74, 142, 149: “Broca’s area” instead of “Broca’s cap”

-Line 73: “in the pre-1.5 Ma Homo specimen” instead of “in pre-1.5 Ma Homo specimen”

-Line 100: we specified “in human brains and endocasts”

-Line 120: “sulcal pattern” instead of “sulcal patterns”

-Line 144: “behaviors” (plural)